# INCREMENT VECTOR TRANSFORMATION FOR CLASS INCREMENTAL LEARNING

## ABSTRACT

Class Incremental Learning (CIL) presents a major challenge due to the phenomenon of catastrophic forgetting. Recent studies on Linear Mode Connectivity (LMC) reveal that Naive-SGD oracle, trained with all historical data, connects to previous task minima through low-loss linear paths—a property generally absent in current CIL methods. In this paper, we explore whether LMC holds for the CIL oracle. Our empirical results confirm the presence of LMC in the CIL oracle, showing that models can retain performance on earlier tasks by following the discovered low-loss linear paths. Motivated by this finding, we propose Increment Vector Transformation (IVT), which leverages the diagonal of the Fisher Information Matrix to approximate Hessian-based transformation, uncovering low-loss linear paths for incremental updates. Our method is orthogonal to existing CIL approaches, serving as a plug-in with minor extra computational costs. Extensive experiments on CIFAR-100, ImageNet-Subset, and ImageNet-Full demonstrate significant performance improvements when integrating IVT with representative CIL methods.

## 1 INTRODUCTION

Class Incremental Learning (CIL) poses a significant challenge in machine learning, requiring models to learn sequentially without access to previous training data. A notorious phenomenon in this paradigm is catastrophic forgetting (McCloskey & Cohen, 1989), where models overwrite previously acquired knowledge when adapting to new tasks. To mitigate this, various approaches have been proposed. *Regularization methods* (Kirkpatrick et al., 2016; Zenke et al., 2017) constrain updates to crucial parameters for past tasks or transfer knowledge from previous tasks through intermediate features and outputs (Kirkpatrick et al., 2016; Hou et al., 2019; Douillard et al., 2020). *Memory replay methods* (Rebuffi et al., 2016; Liu et al., 2020; Luo et al., 2023) retain a subset of exemplars from previous tasks for rehearsal, selecting representative samples to optimize memory efficiency. *Dynamic architecture methods* (Liu et al., 2021; Zhou et al., 2022) introduce new network components to accommodate new tasks. However, despite these advancements, incremental models still fall short compared to oracles trained incrementally with access to all historical data.

Recently, key insights into this performance gap have emerged from the studies on mode connectivity in neural networks (Draxler et al., 2018; Garipov et al., 2018; Frankle et al., 2020). Mode connectivity refers to the existence of low-loss paths that connect different minima in the loss landscape. In CIL, Mirzadeh et al. (2021) demonstrated that the Naive-SGD oracle exhibits more favorable linear mode connectivity (LMC), meaning that a simple linear manifold of low error connects the Naive-SGD oracle and the minima of past tasks. Following this linear path results in minimal degradation of performance on past tasks. In contrast, this property generally does not hold for incremental solutions. Beyond the Naive-SGD, Wen et al. (2023) explored mode connectivity for recent advanced CIL approaches, and empirically found that LMC is still absent in these methods.

In this paper, we further investigate the connection between LMC and CIL by addressing a crucial question: *"Does LMC hold for the oracle of a CIL approach?"*. The significance of this question lies in its implications: If the CIL oracle[1] exhibits LMC, then there must be a transformation to uncover this low-loss linear path for the CIL models. Surprisingly, we empirically demonstrate

---

[1] Hereafter, 'CIL oracle' refers to the oracle of a CIL approach.

that LMC indeed exists for CIL models, and traversing these paths allows the model to maintain high performance on earlier tasks. Moreover, we found that the model can effectively acquire new knowledge without disrupting previously learned information along these paths, striking a balanced stability-plasticity trade-off (Mermillod et al., 2013).

The observation above motivates us to propose a method for finding low-loss linear paths. We begin by theoretically analyzing the inaccuracy of incremental methods. Specifically, we define an increment vector $V_t$, representing the linear path from the old model to the incremental model. As illustrated in Fig. 1, our analysis shows that the CIL oracle $\theta_t^*$ can be approximated by adding an increment vector $V_t$, transformed by a matrix $S_t$, to the old model $\theta_{t-1}^*$. The transformation $S_t$ is derived from the Hessian and captures the curvature of the loss landscapes for both old and new tasks, ensuring updates remain within the low-loss region for previous tasks. Building by this insight, we introduce **I**ncrement **V**ector **T**ransformation (IVT). Since computing the full Hessian is impractical for large neural networks, IVT efficiently approximates it by using the diagonal of the Fisher Information Matrix. This approximation retains essential curvature information while greatly reducing computational overhead, making IVT both efficient and seamlessly compatible with existing CIL methods.

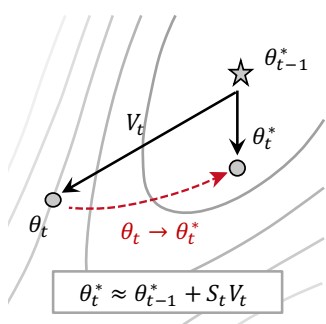

Figure 1: Illustration of IVT. The CIL oracle $\theta_t^*$ can be reached by transforming increment vector $V_t$ of the incremental model $\theta_t$.

Extensive experiments on benchmark datasets, including CIFAR-100, ImageNet-Subset, and ImageNet-Full, demonstrate significant improvements when integrating IVT with existing representative CIL methods. Our contributions are summarized as follows:

- Linear mode connectivity in CIL is empirically analyzed, with a focus on accuracy consistency and the stability-plasticity trade-off along the linear paths.

- A novel method, IVT, is proposed to find low-loss linear paths for CIL, mitigating catastrophic forgetting by transforming the increment vector to a low-loss region for past tasks.

- The effectiveness of IVT is empirically validated on CIFAR-100, ImageNet-Subset, and ImageNet-Full, demonstrating significant performance improvements when integrated with representative CIL methods.

## 2 REVISITING LINEAR MODE CONNECTIVITY IN CIL

The forgetting analysis based on Taylor expansion is commonly used in CIL (Yin et al., 2020; Mirzadeh et al., 2020; Wu et al., 2024). For simplicity, suppose that there are two tasks, $\mathcal{T}_1$ and $\mathcal{T}_2$. Let $\theta_1$ be the minima obtained on $\mathcal{T}_1$, we perform a second-order Taylor expansion of $\mathcal{L}_1(\theta)$ at $\theta_1$:

$$\mathcal{L}_1(\theta) \approx \mathcal{L}_1(\theta_1) + (\theta - \theta_1)^\top \nabla \mathcal{L}_1(\theta_1) + \frac{1}{2}(\theta - \theta_1)^\top H_1(\theta - \theta_1) \tag{1}$$

$$\approx \mathcal{L}_1(\theta_1) + \frac{1}{2}(\theta - \theta_1)^\top H_1(\theta - \theta_1). \tag{2}$$

The last equality holds because, at the minima $\theta_1$ of $\mathcal{T}_1$, the model is assumed to converge and thus $\nabla \mathcal{L}_1(\theta_1) \approx 0$. Besides, the Hessian matrix $H_1 = \nabla^2 \mathcal{L}_1(\theta_1)$ needs to be positive semi-definite at the converged minima. Therefore, the forgetting $F_1$ can be bounded as follows:

$$F_1 = \mathcal{L}_1(\theta) - \mathcal{L}_1(\theta_1) \approx \frac{1}{2}(\theta - \theta_1)^\top H_1(\theta - \theta_1) \leq \frac{1}{2}\lambda^1 \|\Delta\theta\|^2. \tag{3}$$

where $\Delta\theta = \theta - \theta_1$ and $\lambda_1$ is the maximum eigenvalue of $H_1$. When $\Delta\theta$ aligns with the eigenvector corresponding to $\lambda^1$, $F_1$ reaches its upper bound, and the model update follows the direction of maximum curvature of $H_1$. Conversely, reducing $F_1$ can be achieved by minimizing $\Delta\theta$ or by steering the model update direction away from the higher curvature directions of $H_1$.

Recently, some studies have linked catastrophic forgetting in CIL to mode connectivity (Mirzadeh et al., 2020; Verwimp et al., 2021; Wen et al., 2023). Mirzadeh et al. (2021) empirically demonstrate that Naive-SGD oracle obtained through joint training with all previous data lies within the same

low-loss region as the solutions for previous tasks and can be connected to them via low-loss linear paths. Moving along this path does not significantly impact the performance for previous tasks, suggesting that the Naive-SGD oracle has identified low-curvature directions in the loss landscape for earlier tasks. In contrast, this property does not hold for the incremental solution. Moving along the linear path from the previous solution to the incremental solution often results in a substantial drop in accuracy for previous tasks (Mirzadeh et al., 2020; Wen et al., 2023).

Beyond the Naive-SGD, we explore the linear mode connectivity (LMC) for the CIL approaches $\theta_t$ and its oracle $\theta_t^*$, with a particular focus on accuracy consistency and the stability-plasticity trade-off along the linear path. To achieve this, we evaluate the accuracy of a series of interpolation models, starting from the old model $\theta_i^*$ (for $i \leq t - 1$) and progressing along the updated linear direction. Formally, the interpolation models are defined as follows:

$$\bar{\theta}_{t,i}\left(\lambda\right) = \theta_i^* + \lambda U_t. \tag{4}$$

Here, $\lambda$ is the interpolation factor, and $U_t = \left(\theta_t - \theta_i^*\right)/\|\theta_t - \theta_i^*\|_2$ represents the normalized update vector. Similarly, we define the interpolation to the CIL oracle as $\bar{\theta}_{t,i}^*\left(\lambda\right) = \theta_i^* + \lambda U_t^*$, where $U_t^* = \left(\theta_t^* - \theta_i^*\right)/\|\theta_t^* - \theta_i^*\|_2$. Note that adding $U_t$ to $\theta_i^*$ with $\hat{\lambda} = \|\theta_t - \theta_i^*\|_2$ results in $\theta_t$, and adding $U_t^*$ with $\hat{\lambda}^* = \|\theta_t^* - \theta_i^*\|_2$ leads to $\theta_t^*$. For the mismatched parameters between the two interpolated models, e.g., the classifier parameters for the new classes, we initialize them for $\theta_i^*$ as described in (Wen et al., 2023) before interpolation.

## 2.1 ACCURACY CONSISTENCY ALONG THE LINEAR PATH

We evaluate accuracy consistency along the linear path on CIFAR-100 using PODNet (Douillard et al., 2020) and LUCIR (Hou et al., 2019). The experiments consist of an initial task with 50 classes, followed by 5 incremental tasks, each introducing 10 new classes. The incremental model retains 20 exemplars per class, while the CIL oracle has access to the full training data of previous tasks at each incremental step. Fig. 2 illustrates the test accuracy of $\mathcal{T}_1$ along the linear path from $\theta_1^*$ to the models of subsequent tasks, as well as the test accuracy of both $\mathcal{T}_1$ and $\mathcal{T}_2$ as we move from $\theta_2^*$ to the models of later tasks.

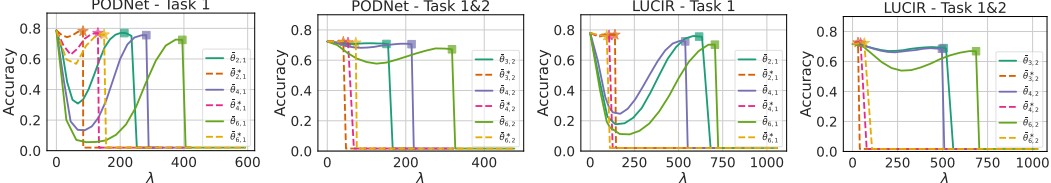

Figure 2: Evaluating accuracy consistency along the linear path on CIFAR-100 for increments of 5 tasks (i.e., 6 tasks in total). The star and square denote the CIL oracle $\theta_t^* = \bar{\theta}^*(\hat{\lambda}^*)$ and the incremental model $\theta_t = \bar{\theta}(\hat{\lambda})$, respectively.

In Fig. 2, we can observe that the CIL oracle achieves better accuracy consistency along the linear path. Concretely, the experiments uncover two key observations: (1) The CIL oracles tend to stay closer to the minima of previous tasks, indicating joint training with old training data prevents the models from moving too far from their previous states, resulting in smaller $\Delta\theta$. (2) The updates of the CIL oracle aligns with the direction of lower curvature. As $\lambda$ increases from 0, the accuracy of $\bar{\theta}$ drops sharply, indicating the presence of a high-loss ridge along the path in the loss landscape. Although the accuracy of $\bar{\theta}$ begins to recover as $\lambda$ continues to increase, it ultimately falls into a sub-optimal basin, as $\bar{\theta}(\hat{\lambda})$ shows significantly lower accuracy compared to $\bar{\theta}(0)$. In contrast, $\bar{\theta}^*$ maintains consistently high accuracy along the linear path, indicating that the CIL oracles remain within the same low-loss basin as the previous minima.

## 2.2 STABILITY-PLASTICITY TRADE-OFF ALONG THE LINEAR PATH

To further investigate the stability-plasticity trade-off of the interpolation models along the linear path, we plot their accuracy on both new and old classes. As depicted in Fig. 3, we interpolate $\theta_1^*$

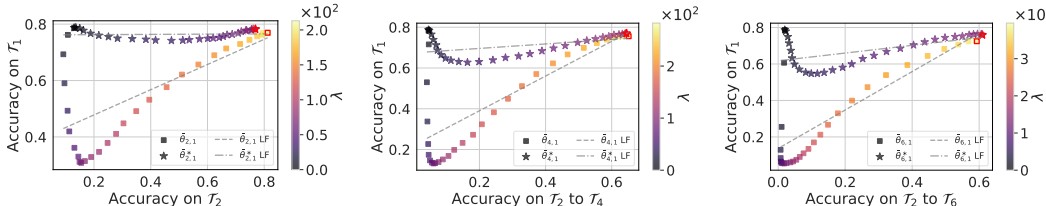

Figure 3: Evaluating stability-plasticity trade-off along the linear path achieved by PODNet on CIFAR-100 for increments of 5 tasks. LT represents the linear fit to the scattered points. The red-edged star and square denote the CIL oracle $\bar{\theta}^*(\hat{\lambda}^*)$ and the incremental model $\bar{\theta}(\hat{\lambda})$, respectively.

with the models of subsequent tasks. The figure reveals that as $\lambda$ increases, $\bar{\theta}^*$ and $\bar{\theta}$ exhibit different behaviors. For $\bar{\theta}$, as $\lambda$ increases from 0 to the midpoint, the accuracy on new classes improves while the accuracy on $\mathcal{T}_1$ drops significantly, highlighting a strong stability-plasticity trade-off. As $\lambda$ continues to increase, $\bar{\theta}$ gradually mitigates this trade-off. In contrast, $\bar{\theta}^*$ demonstrates a more balanced trade-off, maintaining high performance on $\mathcal{T}_1$ while improving accuracy on new classes. This indicates that $\bar{\theta}^*$ effectively integrates new information without significantly compromising previous knowledge. Such behavior suggests that $\bar{\theta}^*$ resides in a more favorable region of the loss landscape, marked by lower curvature and smoother transitions between tasks, allowing it to achieve better overall performance across both old and new classes as $\lambda$ increases.

# 3 APPROACHING ORACLE BY INCREMENT VECTOR TRANSFORMATION

The analysis in Sec. 2 demonstrates the existence of LMC in the CIL oracle. The linear paths discovered by the oracle connect its minima with those of previous tasks while maintaining low loss, providing a promising strategy for addressing catastrophic forgetting in CIL. In this section, we aim to approach the oracle by finding these low-loss linear paths.

Assuming we start from the same old model[2] $\theta_{t-1}^*$, we can express the oracle $\theta_t^*$ and the incremental model $\theta_t$ into the sum of $\theta_{t-1}^*$ and their respective increment vectors $V_t^*$ and $V_t$:

$$\theta_t^* = \theta_{t-1}^* + V_t^*, \quad \theta_t = \theta_{t-1}^* + V_t, \tag{5}$$

where $V_t^* = \theta_t^* - \theta_{t-1}^*$ and $V_t = \theta_t - \theta_{t-1}^*$. Since $V_t^*$ is derived from the joint training with all previous data, obtaining it under the CIL scenario is challenging. However, there should exist a transformation $S_t$ such that:

$$V_t^* = S_t V_t, \quad \theta_t^* = \theta_{t-1}^* + S_t V_t. \tag{6}$$

In other words, we aim to solve for $S_t$ to transform $V_t$ into $V_t^*$, ensuring that the incremental model resides in the low-loss region for previous tasks and remains close to $\theta_{t-1}^*$, as analyzed in Sec. 2.

In what follows, we first theoretically study the inaccuracy of the incremental model and derive the form of $S_t$. We then introduce a practical method that exploits this spirit with almost no additional training cost.

## 3.1 ANALYZING THE INACCURACY OF INCREMENTAL MODEL

We first consider the optimization objectives for the incremental model $\theta_t$ and the oracle $\theta_t^*$ on task $t$. The objective for $\theta_t$ is defined by minimizing the loss function of task $t$, along with a regularization term that approximates the implicit proxy loss of various CIL methods Wu et al. (2024),

$$\theta_t = \arg\min_\theta \mathcal{L}_t(\theta) + \frac{1}{2}\|\theta - \theta_{t-1}^*\|_{\bar{H}_{t-1}}^2, \tag{7}$$

where $\bar{H}_{t-1} = \sum_{i=1}^{t-1} H_i$ is the cumulative Hessian for previous tasks. $\|\Delta\theta\|_{\bar{H}_{t-1}}^2 = \Delta\theta^\top \bar{H}_{t-1}\Delta\theta$ measures how different $\theta$ is from $\theta_{t-1}^*$. The optimization objective for $\theta_t^*$ is similar, but it considers

---

[2]We can also start from $\theta_{t-1}$, which does not affect the derivation.

minimizing the joint loss across all tasks seen up to $t$,

$$\theta_t^* = \arg\min_\theta \sum_{i=1}^{t} \mathcal{L}_i(\theta) + \frac{1}{2}\|\theta - \theta_{t-1}^*\|_{\bar{H}_{t-1}}^2. \tag{8}$$

Based on these optimization objectives, we can quantify the error between $\theta_t$ and $\theta_t^*$ and derive the form of transformation matrix $S_t$ as presented in Proposition 1. For a detailed derivation, please refer to the proof in the Appendix 7.2.

**Proposition 1.** *Consider the incremental model $\theta_t$ and oracle $\theta_t^*$, both initialized from the old model $\theta_{t-1}^*$, with optimization objectives defined in Eqs. 7 and 8. If $\theta_i$ and $\theta_i^*$ are searched within the neighborhood set $\bigcup_{i=1}^{t-1} \mathcal{N}_i$, where $\mathcal{N}_i = \{\theta : d(\theta, \hat{\theta}_i) < \delta_i\}$, then $\theta_t^*$ can be approximately expressed as the sum of $\theta_{t-1}^*$ and an increment vector $(\theta_t - \theta_{t-1})$ transformed by the term $(\bar{H}_{t-1} + \bar{H}_t)^{-1}\bar{H}_t$, which is shown below:*

$$\theta_t^* \approx \theta_{t-1} + (\bar{H}_{t-1} + \bar{H}_t)^{-1}\bar{H}_t(\theta_t - \theta_{t-1}) \tag{9}$$

From the results in Eq. 9, we have the following observations: (1) When $\theta_t$ resides within a relatively flat loss landscape for the old tasks, characterized by a small $\bar{H}_{t-1}$, the approximation indicates that $\theta_t^*$ closely aligns with $\theta_t$. This suggests that the incorporation of new tasks does not significantly disrupt the knowledge acquired from previous tasks. (2) When $\theta_t$ lies in a region of low curvature for the new task, that is, when $H_t$ is small and $\bar{H}_t$ is approximately equal to $\bar{H}_{t-1}$, then $\theta_t^*$ can be approximated as the arithmetic mean of $\theta_t$ and $\theta_{t-1}$.

### 3.2 INCREMENT VECTOR TRANSFORMATION FOR CIL

In neural networks with numerous parameters, explicitly computing the full Hessian matrix is often impractical. The Fisher Information Matrix (FIM) (Fisher, 1922; Amari, 1996) is an efficient alternative for Hessian estimation, as it can be directly derived from first-order derivatives. Building on Proposition 1, we propose a novel method for CIL named Increment Vector Transformation (IVT), which utilizes the diagonal of the FIM.

As is common in existing approaches (Kirkpatrick et al., 2016; Matena & Raffel, 2022; Daheim et al., 2023), we can reduce the computation cost by using the diagonal of the FIM, bringing it to a level comparable to training on $N$ samples. The diagonal of the FIM is computed as follows:

$$F_t = \mathbb{E}_{(x,y)\in\mathcal{T}_t}\left(\nabla\mathcal{L}_t(x,y)\right)^2. \tag{10}$$

In our implementation, we compute the diagonal of FIM in an online manner by accumulating the backpropagated gradients from each batch during training, leading to negligible computational cost. By replacing the Hessian in Eq. 9 with Eq. 10, we formally define IVT as follows:

$$\hat{\theta}_t := \theta_{t-1} + \frac{\bar{F}_t}{\bar{F}_{t-1} + \bar{F}_t}(\theta_t - \theta_{t-1}), \tag{11}$$

where $\bar{F}_t = \sum_{i=1}^{t} F_i$ represents the cumulative diagonal of the FIM up to task $t$. The operation in Eq. 11 consists of simple matrix operations on parameters, performed only at intervals of several epochs. Consequently, IVT is simple, incurs minor extra computational cost, and can be implemented with just a few lines of PyTorch code. It can used as a plug-in to enhance the efficacy of many advanced CIL methods. Algo. 1 presents the pseudo code for IVT.

## 4 EXPERIMENT

We conduct extensive experiments on CIFAR-100 (Krizhevsky et al., 2009), ImageNet-Subset, and ImageNet-Full (Deng et al., 2009). The protocol follows Douillard et al. (2020), where the initial task includes half of the classes, and the remaining classes are evenly distributed across the subsequent incremental tasks, *e.g.*, CIFAR-100 starts with 50 classes, with the remaining classes divided equally over 5, 10, or 25 incremental learning steps. The class order is randomized using seed 1993 (Rebuffi et al., 2016). Our evaluation is consistent with most existing work, using the average incremental accuracy, denoted as $AA = \frac{1}{N}\sum_{t=1}^{T} a_t$, and the last accuracy, $LA = a_T$, where

$a_t$ represents the accuracy over all classes seen after task $t$. To assess forgetting, we use the forgetting measure (Chaudhry et al., 2018), defined as $FM = \frac{1}{T-1} \sum_{i=1}^{T-1} \max_{t \in \{i, T-1\}} (a_{t,i} - a_{T,i})$, with $a_{t,i}$ representing the accuracy of task $i$ after training task $t$.

**Implementation Details.** We conduct extensive experiments on CIFAR-100 (Krizhevsky et al., 2009), ImageNet-Subset, and ImageNet-Full (Deng et al., 2009). We use ResNet-32 (He et al., 2016) with stride 8 for CIFAR-100 and ResNet-18 (He et al., 2016) with stride 32 for both ImageNet-Subset and ImageNet-Full. The optimizer used is SGD, starting with an initial learning rate of 0.1, which decays according to a cosine annealing schedule. On CIFAR-100, we train for 160 epochs, while on ImageNet-Subset and ImageNet-Full, training is conducted for 90 epochs. The batch size is set to 128 across all datasets. The interval for IVT is set to 10 epochs. Unless otherwise specified, the exemplar size is fixed at 20 exemplars per class in all experiments.

---

**Algorithm 1** Increment Vector Transformation (IVT)

1: Train $\theta_1$ on in $\mathcal{T}_1$
2: Compute $F_1$ on $\mathcal{T}_1$ by Eq. 10
3: **for** incremental task $\mathcal{T}_t \in \{\mathcal{T}_2, \mathcal{T}_3, \cdots\}$ **do**
4:     Initialize $\theta_t \leftarrow \theta_{t-1}$
5:     **for** Epoch $\in \{1, 2, \cdots\}$ **do**
6:        Initialize $F_t = \mathbf{0}$
7:        **for** mini-batch $\mathcal{B}_i \in$ permute($\{\mathcal{B}_1, \mathcal{B}_2, \cdots\}$) **do**
8:           Compute $g_i = \mathbb{E}_{(x,y) \in \mathcal{B}_i} (\nabla \mathcal{L}_t(x, y))$
9:           Update $\theta_t \leftarrow$ CIL Method$(\theta_t, g_i)$
10:        **end for**
11:        Compute $F_t = \mathbb{E}_i(g_i^2)$
12:        **if** Epoch mod Interval $= 0$ **then**
13:           Update $\theta_t \leftarrow \theta_{t-1} + \frac{\bar{F}_t}{\bar{F}_{t-1} + \bar{F}_t}(\theta_t - \theta_{t-1})$
14:        **end if**
15:     **end for**
16: **end for**

---

**Comparison Methods.** Our method (IVT) is orthogonal to existing CIL approaches and can augment their efficacy as a plug-in unit. We select PODNet (Douillard et al., 2020) and AFC (Kang et al., 2022) as representative methods for adapting IVT. For comparison, we use iCaRL (Rebuffi et al., 2016), BiC (Wu et al., 2019), LUCIR (Hou et al., 2019), Mnemonics (Liu et al., 2020), GeoDL (Simon et al., 2021), and EOPC (Wen et al., 2023) as our baseline methods.

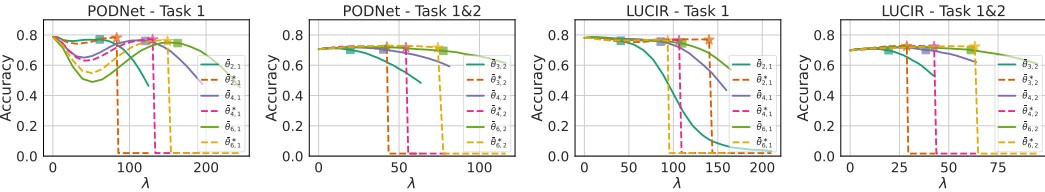

Figure 4: Evaluating accuracy consistency along the linear path on CIFAR-100 for increments of 5 tasks. The star denotes the CIL oracle $\bar{\theta}^*(\hat{\lambda}^*)$ and square denotes the IVT model $\bar{\theta}(\hat{\lambda})$.

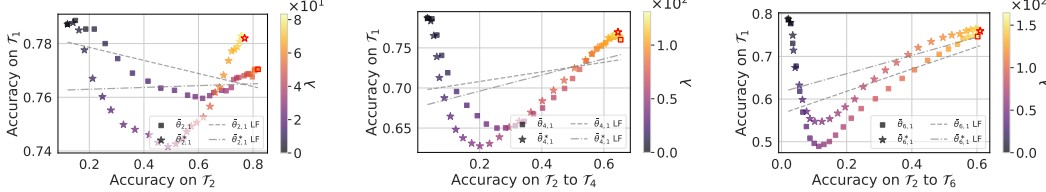

Figure 5: Evaluating stability-plasticity trade-off achieved by PODNet along the linear path on CIFAR-100 for increments of 5 tasks. LT represents the linear fit to the scattered points. The red-edged star denotes the CIL oracle $\bar{\theta}^*(\hat{\lambda}^*)$ and square denotes the IVT model $\bar{\theta}(\hat{\lambda})$.

### 4.1 ANALYTICAL EXPERIMENTS

**Linear Mode Connectivity along the Linear Path.** Similar to Sec. 2, we analyze the LMC of the IVT model. As shown in Fig. 4, the IVT model demonstrates LMC behavior comparable to the CIL oracle along the linear path. As $\lambda$ increases, the IVT model experiences only a slight accuracy decline, while its distance to the old model remains closely aligned with that of the oracle. This suggests that both the IVT model and the oracle occupy low-curvature regions in the loss landscape for old tasks, staying close to the old model. Moreover, Fig. 5 illustrates that the IVT model achieves

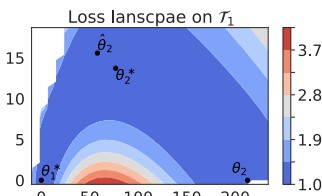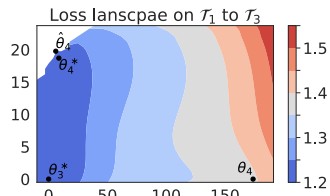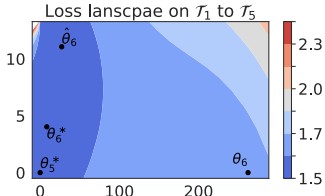

Figure 6: Visualization of the training loss landscape in parameter vector space, produced by PODNet on CIFAR-100 with increments of 5 tasks.

a stability-plasticity trade-off comparable to the oracle. In comparison to Fig. 3, IVT significantly mitigates this trade-off, allowing it to acquire new tasks with minimal interference to previously learned knowledge.

**Training Loss Landscape.**  To better understand the relationships between the old model $\theta_{t-1}^*$, the incremental model $\theta_t$, the IVT model $\hat{\theta}_t$, and the oracle $\theta_t^*$, we visualize the training loss landscape in the parameter vector space, following (Mirzadeh et al., 2021). As shown in Fig. 6, the IVT model $\hat{\theta}_t$ stays closer to $\theta_t^*$ compared to the incremental model $\theta_t$. The visualization illustrates that $\theta_{t-1}^*$, $\hat{\theta}_t$, and $\theta_t^*$ all reside within the same low-loss region, allowing the model to maintain strong performance on previously learned tasks. In contrast, the incremental model $\theta_t$ drifts into regions with higher loss, indicating difficulties in retaining knowledge from prior tasks. This observation supports the effectiveness of IVT in guiding model updates to remain within the low-loss region of earlier tasks, thus mitigating catastrophic forgetting and promoting stability during incremental learning.

**The Effect of IVT Interval.**  The stationarity condition is provided in Proposition 1. In general, a short interval leads to inaccurate transformations, while a long interval reduces the chance of finding a low-loss linear path. Therefore, selecting an appropriate interval is crucial. We conduct sensitivity experiments on the interval, the only hyperparameter of IVT. As shown in Fig. 7, IVT is robust to interval variations and consistently improves baseline performance.

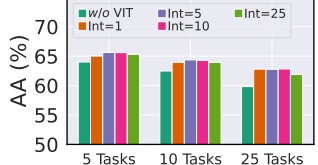

Figure 7: Ablating IVT interval with PODNet on CIFAR-100.

Table 1: Ablating exemplar size $|\mathcal{E}|$ on CIFAR-100 with increments of 5 tasks.

| Method | $|\mathcal{E}| = 5$ | | | $|\mathcal{E}| = 10$ | | | $|\mathcal{E}| = 20$ | | |
| | $AA\uparrow$ | $LA\uparrow$ | $FM\downarrow$ | $AA\uparrow$ | $LA\uparrow$ | $FM\downarrow$ | $AA\uparrow$ | $LA\uparrow$ | $FM\downarrow$ |
|---|---|---|---|---|---|---|---|---|---|
| PODNet | 54.63 | 45.50 | 26.06 | 61.28 | 49.87 | 21.82 | 64.00 | 56.47 | 17.72 |
| *w/* EOPC | 56.52 | 46.82 | 14.20 | 63.96 | 52.54 | 12.37 | 65.36 | 55.55 | 8.45 |
| *w/* IVT | 62.04 | 51.33 | 17.11 | 63.02 | 53.59 | 13.06 | 65.36 | 56.61 | 11.68 |

Table 2: Training time (s) for each incremental task.

| Method | CIFAR-100 | | |
| | 5 | 10 | 25 |
|---|---|---|---|
| PODNet | 621 | 487 | 326 |
| *w/* IVT | 655 | 495 | 396 |

**The Effect of Exemplar Size.**  We investigate the effect of IVT on exemplar size and compare it to EOPC. EOPC leverages exemplars to identify low-loss paths, typically resulting in a nonlinear optimized trajectory. In contrast, our method does not rely on exemplars but instead uses the diagonal of the FIM. As shown in Tab. 1, IVT consistently improves baseline performance, particularly in the low-exemplar regime. When sufficient exemplars are available, IVT achieves results comparable to EOPC. This highlights the effectiveness of IVT and its robustness in scenarios with limited exemplars.

**Time Complexity.**  To investigate whether IVT introduces extra computational overhead when adapted to CIL methods, we conducted a time complexity analysis. As shown in Tab. 2, IVT results in only a slight increase in training time compared to the baseline methods. The experiments demonstrate that our method ensures high computational efficiency.

Table 3: Comparative results (%) on CIFAR-100 with different numbers of incremental tasks. The results are averaged over 3 random runs, with both the mean and standard deviation reported. Results marked with † and ‡ are referenced from (Simon et al., 2021) and (Wen et al., 2023), respectively. ⋆ indicates reproduced EOPC+PODNet results.

| Method | 5 Tasks | | | 10 Tasks | | | 25 Tasks | | |
|---|---|---|---|---|---|---|---|---|---|
| | $AA\uparrow$ | $LA\uparrow$ | $FM\downarrow$ | $AA\uparrow$ | $LA\uparrow$ | $FM\downarrow$ | $AA\uparrow$ | $LA\uparrow$ | $FM\downarrow$ |
| iCaRL‡ | 57.83 | – | 25.16 | 52.63 | – | 26.57 | 49.02 | – | 29.83 |
| BiC† | 59.36 | 49.56 | – | 54.20 | 45.28 | – | 50.00 | – | – |
| LUCIR‡ | 63.62 | – | 19.58 | 60.95 | – | 19.79 | 57.79 | – | 20.31 |
| Mnemonics† | 63.34 | 52.14 | – | 62.28 | 52.53 | – | 60.96 | – | – |
| GeoDL† | 65.14 | 55.62 | – | **65.03** | 55.26 | – | 63.12 | – | – |
| EOPC⋆ | 65.36 | 55.55 | 8.45 | 63.44 | 53.88 | **8.68** | 61.44 | 51.27 | **11.29** |
| PODNet | 64.00$_{(\pm0.54)}$ | 54.47$_{(\pm0.88)}$ | 17.72$_{(\pm0.27)}$ | 62.47$_{(\pm0.51)}$ | 52.89$_{(\pm0.80)}$ | 21.57$_{(\pm0.38)}$ | 59.82$_{(\pm0.84)}$ | 50.71$_{(\pm0.96)}$ | 25.90$_{(\pm0.89)}$ |
| *w/* IVT | 65.36$_{(\pm0.24)}$ | 56.61$_{(\pm0.47)}$ | 11.68$_{(\pm0.47)}$ | 63.45$_{(\pm0.72)}$ | 55.41$_{(\pm0.72)}$ | 12.87$_{(\pm0.47)}$ | 61.74$_{(\pm0.98)}$ | 53.43$_{(\pm1.15)}$ | 15.84$_{(\pm0.76)}$ |
| AFC | 65.51$_{(\pm0.33)}$ | 56.25$_{(\pm0.54)}$ | 11.16$_{(\pm0.56)}$ | 64.00$_{(\pm0.77)}$ | 54.37$_{(\pm0.83)}$ | 14.31$_{(\pm0.46)}$ | 62.53$_{(\pm0.68)}$ | 53.86$_{(\pm0.86)}$ | 17.90$_{(\pm0.38)}$ |
| *w/* IVT | **65.94**$_{(\pm0.32)}$ | **56.62**$_{(\pm0.59)}$ | **8.44**$_{(\pm0.39)}$ | 64.53$_{(\pm0.64)}$ | **56.00**$_{(\pm1.25)}$ | 10.00$_{(\pm0.53)}$ | **63.36**$_{(\pm0.74)}$ | **54.77**$_{(\pm0.81)}$ | 14.05$_{(\pm0.30)}$ |

Table 4: Comparative results (%) on ImageNet-Subset and ImageNet-Full with different numbers of incremental tasks. Results marked with † and ‡ are referenced from (Simon et al., 2021) and (Wen et al., 2023), respectively.

| Method | ImageNet-Subset | | | | | | | | | ImageNet-Full | | |
|---|---|---|---|---|---|---|---|---|---|---|---|---|
| | 5 Tasks | | | 10 Tasks | | | 25 Tasks | | | 10 Tasks | | |
| | $AA\uparrow$ | $LA\uparrow$ | $FM\downarrow$ | $AA\uparrow$ | $LA\uparrow$ | $FM\downarrow$ | $AA\uparrow$ | $LA\uparrow$ | $FM\downarrow$ | $AA\uparrow$ | $LA\uparrow$ | $FM\downarrow$ |
| iCaRL‡ | 64.75 | – | 24.22 | 58.80 | – | 29.63 | 52.46 | – | 32.58 | 47.42 | – | 15.94 |
| BiC† | 70.07 | 60.34 | – | 64.96 | 56.18 | – | 57.73 | – | – | 58.72 | 51.23 | – |
| LUCIR‡ | 71.93 | – | 20.56 | 69.43 | – | 25.97 | 63.51 | – | 28.55 | 61.63 | – | 26.99 |
| Mnemonics† | 72.58 | 64.58 | – | 71.37 | 62.52 | – | 69.74 | – | – | 63.01 | 55.45 | – |
| GeoDL† | 73.87 | 67.37 | – | 73.55 | 65.57 | – | 71.72 | – | – | 64.46 | 56.75 | – |
| PODNet | 72.41 | 63.06 | 14.04 | 69.69 | 59.28 | 18.38 | 59.10 | 48.04 | 29.56 | 64.10 | 55.57 | 14.09 |
| *w/* IVT | 73.57 | 65.10 | 8.81 | 71.29 | 62.76 | 10.05 | 66.74 | 55.64 | 16.17 | **65.07** | 56.95 | **13.00** |
| AFC | 76.15 | 70.20 | 5.87 | 74.49 | 66.88 | 11.00 | 71.19 | 62.36 | 13.92 | 64.36 | 56.86 | 13.80 |
| *w/* IVT | **76.58** | **70.68** | **3.67** | **74.95** | **67.68** | **7.92** | **72.15** | **63.46** | 13.87 | 64.87 | **57.36** | 13.26 |

## 4.2 COMPARATIVE RESULTS

**Results on CIFAR-100.** To evaluate the effectiveness of IVT, it is applied to two prominent CIL methods, PODNet and AFC. Tab. 3 summarizes the comparative results, demonstrating IVT's significant improvements on CIFAR-100. For PODNet, IVT improves average incremental accuracy by 1.36%, 0.98%, and 1.92% over 5, 10, and 25 steps, respectively. Additionally, the last accuracy is improved by 2.14%, 2.52%, and 2.72%, while the forgetting measure is reduced by 6.04%, 8.70%, and 10.06% across the same steps. IVT also yields substantial performance gains for AFC, notably decreasing forgetting.

**Results on ImageNet.** Tab. 4 further presents the comparative and adaptation results of IVT on both ImageNet-Subset and ImageNet-Full. On ImageNet-Subset, IVT enhances PODNet's average incremental accuracy by 1.16%, 1.60%, and 7.64% across 5, 10, and 25 steps, respectively. Furthermore, the last accuracy is improved by 2.04%, 3.48%, and 7.60%, while the forgetting measure is reduced by 5.23%, 8.33%, and 13.39%. For ImageNet-Full, IVT delivers improvements of 0.96% in average incremental accuracy and 1.38% in last accuracy, reducing the forgetting measure by 1.09%. AFC similarly benefits from IVT, showing enhanced performance and reduced forgetting.

## 5 RELATED WORK

**Class Incremental Learning.** Existing CIL methods can be broadly categorized into three main approaches. *Regularization methods* mitigate catastrophic forgetting by imposing constraints on model parameters or outputs. Approaches like EWC (Kirkpatrick et al., 2016) calculate the importance of parameters for previous tasks and penalize changes to crucial parameters, while knowledge distillation techniques such as LUCIR (Hou et al., 2019), PODNet (Douillard et al., 2020), and GeoDL (Simon et al., 2021) use output logits or intermediate features to preserve learned representations.

To address class imbalance, methods like BiC (Wu et al., 2019) and FOSTER (Wang et al., 2022a) apply post-hoc corrections and classifier adjustments to reduce bias toward newly introduced classes. *Memory replay methods* store a subset of exemplars and replay them during new task learning. For instance, iCaRL (Rebuffi et al., 2016) selects samples that best approximate class means, while Mnemonics (Liu et al., 2020) and CIM (Luo et al., 2023) optimize exemplar selection or compression to maximize memory efficiency. When storing real data is infeasible due to privacy or memory constraints, prompt-based methods (Wang et al., 2022c;b), prototype-based approaches (Zhu et al., 2021; 2022), and synthetic data techniques (Choi et al., 2021; Qiu et al., 2024) simulate replay without violating these constraints. *Dynamic architecture methods* adapt the network structure to accommodate new tasks by expanding or modifying network components. Approaches like AANet (Liu et al., 2021) and MEMO (Zhou et al., 2022) dynamically allocate resources, effectively isolating new knowledge from previously acquired information. This adaptability balances stability and plasticity, allowing the model to learn new information flexibly while preserving existing knowledge.

**Mode Connectivity.** Mode connectivity is a phenomenon where different minima in the loss landscape of deep neural networks are connected by low-loss paths in the parameter space (Draxler et al., 2018; Garipov et al., 2018). It offers a novel perspective on optimization, suggesting that optima obtained through gradient-based methods are points on a connected, low-loss manifold. Various methods, such as polygonal chains, Bézier curves, elastic bands, and simplicial complexes, have been used to model these low-loss paths (Draxler et al., 2018; Garipov et al., 2018; Benton et al., 2021). The initialization of minima plays a crucial role: high-loss ridge often exists along the linear path between minima trained from different initializations, but linear connectivity can be achieved when minima share the same initialization and are stable to SGD noise (Frankle et al., 2020; Neyshabur et al., 2020). Mode connectivity advances our understanding of neural network optimization and facilitates applications in loss landscape analysis, weight pruning, and model ensembling (Draxler et al., 2018; Frankle et al., 2020; Fort & Jastrzebski, 2019).

## 6 CONCLUSION

In this paper, we investigate whether LMC holds in the CIL oracle and confirm that models can retain performance on earlier tasks by following these low-loss linear paths. Inspired by this finding, we introduce Increment Vector Transformation (IVT), a method that uses the diagonal of the Fisher Information Matrix to approximate a Hessian-based transformation, allowing the discovery of low-loss linear paths for incremental updates. IVT is compatible with existing CIL methods and requires minimal additional computational overhead. Extensive experiments on CIFAR-100, ImageNet-Subset, and ImageNet-Full demonstrate that integrating IVT with state-of-the-art CIL methods leads to substantial performance improvements.

**Limitations.** Since IVT is a transformation method based on Hessian information, the accuracy of Hessian estimation is critical. Our use of the diagonal Fisher Information Matrix approximation may not achieve high accuracy. Furthermore, as tasks progress, the effectiveness of the accumulated diagonal Fisher Information Matrix stored by IVT may decrease. Updating the Hessian information for past tasks is likely to improve performance. We leave these considerations for future work.

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

# 7 APPENDIX

## 7.1 THE DETAILS OF BENCHMARK DATASETS

To achieve a comprehensive study, we conduct extensive experiments in the main paper, including datasets CIFAR-100 (Krizhevsky et al., 2009), ImageNet-Subset (Deng et al., 2009), and ImageNet-Full (Deng et al., 2009).

- CIFAR-100 is a widely-used image classification dataset, consisting of 60,000 color images with dimensions of 32×32 pixels, across 100 different classes. Each class in the dataset is designed to represent a distinct object category (*e.g.*, animals, vehicles, everyday objects). The dataset is split into a training set of 50,000 images, with 500 images per class, and a validation (or test) set of 10,000 images, with 100 images per class.

- ImageNet-subset (ImageNet-100) is a smaller, 100-class subset derived from the larger ImageNet dataset. It is frequently used for tasks like transfer learning and incremental learning, offering a balance between dataset size and complexity. Each class in ImageNet-Subset contains approximately 1,300 training images and 50 validation images, making it a more computationally manageable version of the full ImageNet dataset while still providing substantial class diversity and variability in visual content.

- ImageNet-Full (ImageNet-1000) refers to the subset of ImageNet containing 1,000 classes. It is the most commonly used version of ImageNet for tasks such as image classification, pretraining, and benchmarking deep learning models. This dataset includes around 1.2 million training images and 50,000 validation images, with approximately 50 images per class in the validation set. Each class in ImageNet-full represents a distinct object category, ranging from animals to everyday objects.

## 7.2 PROOF PROPOSITION 1

**Proposition 2.** *Consider the incremental model $\theta_t$ and oracle $\theta_t^*$, both initialized from the old model $\theta_{t-1}^*$, with optimization objectives defined in Eqs. 7 and 8. If $\theta_i$ and $\theta_i^*$ are searched within the neighborhood set $\bigcup_{i=1}^{t-1} \mathcal{N}_i$, where $\mathcal{N}_i = \{\theta : d(\theta, \hat{\theta}_i) < \delta_i\}$, then $\theta_t^*$ can be approximately expressed as the sum of $\theta_{t-1}^*$ and an increment vector $(\theta_t - \theta_{t-1})$ transformed by the term $(\bar{H}_{t-1} + \bar{H}_t)^{-1} \bar{H}_t$, which is shown below:*

$$\theta_t^* \approx \theta_{t-1} + (\bar{H}_{t-1} + \bar{H}_t)^{-1} \bar{H}_t (\theta_t - \theta_{t-1})$$

*Proof.* We begin by stating the stationarity conditions for both the incremental model $\theta_t$ and the oracle $\theta_t^*$, which are derived from setting the derivatives of the objectives in Eqs. 7 and 8 to zero:

$$\bar{H}_{t-1}(\theta_t - \theta_{t-1}) = -\nabla \mathcal{L}_t(\theta_t), \tag{12}$$

$$\bar{H}_{t-1}(\theta_t^* - \theta_{t-1}) = -\sum_{i=1}^{t} \nabla \mathcal{L}_i(\theta_t^*), \tag{13}$$

Next, we subtract Eq. 13 from Eq. 12, yielding:

$$\bar{H}_{t-1}(\theta_t^* - \theta_t) = -\sum_{i=1}^{t-1} \nabla \mathcal{L}_i(\theta_t^*) - [\nabla \mathcal{L}_t(\theta_t^*) - \nabla \mathcal{L}_t(\theta_t)]. \tag{14}$$

To proceed, we apply a first-order Taylor approximation to approximate the difference between the gradients:

$$\nabla \mathcal{L}_t(\theta_t^*) - \nabla \mathcal{L}_t(\theta_t) = H_t(\theta_t^* - \theta_t). \tag{15}$$

Substituting Eq. 15 into Eq. 14, we obtain:

$$\bar{H}_{t-1}(\theta_t^* - \theta_t) = -\sum_{i=1}^{t-1} \nabla \mathcal{L}_i(\theta_t^*) - H_t(\theta_t^* - \theta_t). \tag{16}$$

We then move the term $H_t(\theta_t^* - \theta_t)$ to the left-hand side and multiply the entire expression by $\bar{H}_t^{-1}$:

$$\theta_t^* - \theta_t = -\bar{H}_t^{-1} \sum_{i=1}^{t-1} \nabla \mathcal{L}_i(\theta_t^*) \tag{17}$$

Now, by approximating $\theta_i^*$ for each $i$ as in Eq. 15, we express:

$$\theta_t^* - \theta_t = -\bar{H}_t^{-1} \sum_{i=1}^{t-1} [\nabla \mathcal{L}_i(\theta_i) + H_i(\theta_t^* - \theta_i)] \tag{18}$$

Since the gradient $\nabla \mathcal{L}i$ is close to zero for the converged old model, it can be neglected in practice, leading to:

$$\theta_t^* - \theta_t \approx -\bar{H}_t^{-1} \sum_{i=1}^{t-1} H_i(\theta_t^* - \theta_i) \tag{19}$$

Assuming the parameters are searched within the neighborhood set $\bigcup_{i=1}^{t-1} \mathcal{N}_i$, where $\mathcal{N}_i = \{\theta : d(\theta, \hat{\theta}_i) < \delta_i\}$, we follow the approximation from (Huszár, 2018):

$$\sum_{i=1}^{j-1} H_i(\theta - \theta_i) \approx (\sum_{i=1}^{j-1} H_i)(\theta - \theta_{j-1}) \tag{20}$$

Substituting Eq. 20 into Eq. 19 and rearranging with respect to $\theta_t$, we recover Eq. 9:

$$\theta_t^* \approx \theta_{t-1} + (\bar{H}_{t-1} + \bar{H}_t)^{-1} \bar{H}_t(\theta_t - \theta_{t-1}) \tag{21}$$

$\square$

### 7.3 DETAILED COMPARATIVE RESULTS

For a fair comparison with subsequent work, we provide the detailed comparative results in Tab. 3 and Tab. 4.

Table 5: Classification accuracy (%) on CIFAR-100 for 5 increments.

| Method | Step | | | | | |
|---|---|---|---|---|---|---|
| | 1 | 2 | 3 | 4 | 5 | 6 |
| PODNet | 79.56 | 69.726 | 65.25 | 60.22 | 54.74 | 54.47 |
| PODNet *w/* IVT | 79.56 | 70.80 | 66.15 | 61.82 | 57.22 | 56.62 |
| AFC | 79.71 | 71.57 | 67.09 | 62.00 | 56.44 | 56.24 |
| AFC *w/* IVT | 79.71 | 71.74 | 67.13 | 62.54 | 57.90 | 56.62 |

Table 6: Classification accuracy (%) on CIFAR-100 for 10 increments.

| Method | Step | | | | | | | | | | |
|---|---|---|---|---|---|---|---|---|---|---|---|
| | 1 | 2 | 3 | 4 | 5 | 6 | 7 | 8 | 9 | 10 | 11 |
| PODNet | 79.56 | 73.89 | 68.45 | 64.94 | 63.30 | 60.97 | 58.72 | 56.96 | 53.46 | 53.97 | 52.89 |
| PODNet *w/* IVT | 78.76 | 72.99 | 68.83 | 65.43 | 64.12 | 62.15 | 59.86 | 58.92 | 55.48 | 55.99 | 55.41 |
| AFC | 79.71 | 74.49 | 70.05 | 67.01 | 65.48 | 63.52 | 60.86 | 58.57 | 54.59 | 55.30 | 54.37 |
| AFC *w/* IVT | 79.71 | 74.49 | 69.84 | 66.94 | 65.83 | 63.48 | 61.12 | 59.16 | 56.38 | 56.83 | 55.99 |

Table 7: Classification accuracy (%) on CIFAR-100 for 25 increments.

| Method | | | | | | *Step* | | | | | | | | |
|---|---|---|---|---|---|---|---|---|---|---|---|---|---|---|
| | 1 | 2 | 3 | 4 | 5 | 6 | 7 | 8 | 9 | 10 | 11 | 12 | 13 | 14 |
| PODNet | 79.56 | 74.62 | 72.46 | 70.65 | 66.82 | 65.49 | 63.76 | 62.17 | 61.25 | 60.51 | 60.57 | 59.74 | 59.42 | 57.72 |
| PODNet *w/* IVT | 79.56 | 74.73 | 73.37 | 71.47 | 67.72 | 66.61 | 65.45 | 64.11 | 63.16 | 62.45 | 62.20 | 61.08 | 60.97 | 59.82 |
| AFC | 79.71 | 75.81 | 74.59 | 72.79 | 69.03 | 67.86 | 67.33 | 66.04 | 64.60 | 64.30 | 63.53 | 62.61 | 62.34 | 60.95 |
| AFC *w/* IVT | 79.71 | 76.06 | 74.64 | 72.90 | 69.45 | 68.63 | 67.93 | 66.66 | 65.30 | 65.08 | 64.12 | 63.13 | 63.21 | 62.16 |

| Method | | | | | | *Step* | | | | | | |
|---|---|---|---|---|---|---|---|---|---|---|---|---|
| | 15 | 16 | 17 | 18 | 19 | 20 | 21 | 22 | 23 | 24 | 25 | 26 |
| PODNet | 57.07 | 55.95 | 55.77 | 54.98 | 54.37 | 54.12 | 51.16 | 52.38 | 51.75 | 51.34 | 50.92 | 50.71 |
| PODNet *w/* IVT | 59.47 | 58.77 | 58.29 | 57.16 | 57.30 | 56.69 | 53.83 | 55.02 | 54.68 | 53.95 | 53.81 | 53.43 |
| AFC | 59.39 | 58.95 | 58.28 | 57.02 | 56.84 | 56.62 | 54.33 | 55.15 | 54.92 | 54.64 | 54.27 | 53.86 |
| AFC *w/* IVT | 60.37 | 60.21 | 59.40 | 58.02 | 58.26 | 57.83 | 55.68 | 56.51 | 56.14 | 55.74 | 55.36 | 54.77 |

Table 8: Classification accuracy (%) on ImageNet-Subset for 5 increments.

| Method | | | *Step* | | | |
|---|---|---|---|---|---|---|
| | 1 | 2 | 3 | 4 | 5 | 6 |
| PODNet | 84.60 | 78.00 | 72.49 | 70.47 | 65.82 | 63.06 |
| PODNet *w/* IVT | 84.60 | 78.67 | 73.66 | 71.88 | 67.53 | 65.10 |
| AFC | 83.60 | 80.43 | 77.14 | 74.70 | 70.84 | 70.20 |
| AFC *w/* IVT | 83.60 | 80.53 | 77.69 | 75.22 | 71.76 | 70.68 |

Table 9: Classification accuracy (%) on ImageNet-Subset for 10 increments.

| Method | | | | | *Step* | | | | | | |
|---|---|---|---|---|---|---|---|---|---|---|---|
| | 1 | 2 | 3 | 4 | 5 | 6 | 7 | 8 | 9 | 10 | 11 |
| PODNet | 84.64 | 80.11 | 74.63 | 72.28 | 70.31 | 69.39 | 67.95 | 65.20 | 62.18 | 60.63 | 59.28 |
| PODNet *w/* IVT | 84.60 | 80.58 | 76.73 | 74.06 | 71.09 | 70.43 | 68.97 | 66.42 | 64.56 | 63.98 | 62.76 |
| AFC | 83.84 | 82.00 | 78.47 | 77.11 | 75.17 | 74.03 | 73.00 | 70.64 | 69.22 | 68.99 | 66.88 |
| AFC *w/* IVT | 83.60 | 83.02 | 78.77 | 76.83 | 75.46 | 75.15 | 73.52 | 71.53 | 69.87 | 69.07 | 67.68 |

Table 10: Classification accuracy (%) on ImageNet-Subset for 25 increments.

| Method | | | | | | *Step* | | | | | | | | |
|---|---|---|---|---|---|---|---|---|---|---|---|---|---|---|
| | 1 | 2 | 3 | 4 | 5 | 6 | 7 | 8 | 9 | 10 | 11 | 12 | 13 | 14 |
| PODNet | 84.60 | 72.69 | 69.89 | 68.68 | 65.17 | 64.43 | 63.03 | 61.78 | 60.97 | 59.06 | 57.80 | 58.17 | 58.41 | 58.21 |
| PODNet *w/* IVT | 84.60 | 80.04 | 78.41 | 77.25 | 74.17 | 73.80 | 72.03 | 71.41 | 69.36 | 67.18 | 64.91 | 66.94 | 65.95 | 64.97 |
| AFC | 83.60 | 80.65 | 80.78 | 80.04 | 78.48 | 75.70 | 75.61 | 73.94 | 72.88 | 72.47 | 72.94 | 71.50 | 72.14 | 70.97 |
| AFC *w/* IVT | 83.60 | 83.00 | 81.81 | 81.32 | 79.90 | 76.77 | 76.55 | 74.94 | 74.36 | 73.00 | 73.69 | 71.56 | 72.22 | 71.92 |

| Method | | | | | | *Step* | | | | | | |
|---|---|---|---|---|---|---|---|---|---|---|---|---|
| | 15 | 16 | 17 | 18 | 19 | 20 | 21 | 22 | 23 | 24 | 25 | 26 |
| PODNet | 57.85 | 56.63 | 56.10 | 54.07 | 54.63 | 53.93 | 51.80 | 51.39 | 50.68 | 49.83 | 48.88 | 48.04 |
| PODNet *w/* IVT | 65.05 | 63.90 | 63.95 | 62.24 | 61.12 | 61.50 | 60.24 | 59.26 | 57.91 | 57.04 | 56.49 | 55.64 |
| AFC | 69.82 | 69.37 | 67.76 | 67.74 | 66.47 | 66.41 | 64.67 | 65.26 | 63.66 | 63.02 | 62.65 | 62.36 |
| AFC *w/* IVT | 70.23 | 70.00 | 69.17 | 68.24 | 67.47 | 67.36 | 66.18 | 66.43 | 64.83 | 64.44 | 63.41 | 63.46 |

Table 11: Classification accuracy (%) on ImageNet-Full for 10 increments.

| Method | | | | | *Step* | | | | | | |
|---|---|---|---|---|---|---|---|---|---|---|---|
| | 1 | 2 | 3 | 4 | 5 | 6 | 7 | 8 | 9 | 10 | 11 |
| PODNet | 76.83 | 72.85 | 69.68 | 67.20 | 64.72 | 62.87 | 61.10 | 59.52 | 57.96 | 56.80 | 55.57 |
| PODNet *w/* IVT | 76.91 | 73.16 | 70.43 | 68.04 | 65.60 | 63.79 | 62.23 | 60.97 | 59.48 | 58.25 | 56.95 |
| AFC | 76.82 | 72.02 | 69.21 | 67.06 | 64.91 | 63.16 | 61.32 | 60.18 | 58.74 | 57.71 | 56.86 |
| AFC *w/* IVT | 76.81 | 72.28 | 69.73 | 67.53 | 65.19 | 63.46 | 62.10 | 60.97 | 59.55 | 58.55 | 57.36 |