# OpenReview forum: "Increment Vector Transformation for Class Incremental Learning"
_ICLR.cc/2025/Conference — ICLR 2025 Conference Withdrawn Submission_

### Official Review · Reviewer_Yw8m · 2024-10-29

**Soundness:** 3
**Presentation:** 3
**Contribution:** 3
**Rating:** 5
**Confidence:** 4

**Summary:**

The paper explores the Linear Mode Connectivity (LMC) for CIL tasks. This LMC usually does not hold in CIL tasks in previous works, and is revealed in this paper. The empirical results confirm the presence of LMC in the CIL oracle. Hence, a simple manipulation of Increment
Vector Transformation (IVT) can help largely reduce the catastrophic forgetting. This method leverages the diagonal of the Fisher Information Matrix to approximate Hessian-based transformation (though not lossless), uncovering low-loss linear paths for incremental updates, and achieves workable results by inserting the IVT into several methods as a plug-in.

**Strengths:**

1. The way of delivering the paper is very good.
2. The motivation in this paper is very clear and sound.
3. The proposed method is simple and easy to follow.

**Weaknesses:**

1. Baselines are quite old (though 2023 ones are just about 1 years old). Please include more works in 2024, and there should be enough  candidates.
2. Please indicates clearly if this is a training from scratch or training from half scenario.
3. Literature is weak. Quite a few number of recent works are not mentioned. For instance, "Online Hyperparameter Optimization for Class-Incremental Learning" and its series of works, and "DS-AL: A dual-stream analytic learning for exemplar-free class-incremental learning" and its series of works.
4. To be honest, the definition of "oracle" is not clear to everyone in the area of CIL. There is no clear explanation in the manuscript. Looks like a jargon.
5. My largest concern: the performance improvement is not very handsome or consistent across various embeded methods. For instance, the improvement upon AFC is trivial in many cases. And this is the one the authors put on paper.

**Questions:**

See weaknesses.

---

### Official Review · Reviewer_AeMD · 2024-10-29

**Soundness:** 3
**Presentation:** 3
**Contribution:** 2
**Rating:** 3
**Confidence:** 4

**Summary:**

This paper introduces Increment Vector Transformation (IVT) as a method for Class Incremental Learning (CIL) to reduce catastrophic forgetting. By utilizing Linear Mode Connectivity (LMC), the authors propose IVT as a plug-in solution that identifies low-loss linear paths, allowing models to retain performance on previous tasks. The method approximates a Hessian-based transformation using the diagonal of the Fisher Information Matrix, which keeps computational costs low. Experiments on CIFAR-100, ImageNet-Subset, and ImageNet-Full demonstrate the effectiveness of IVT.

**Strengths:**

+ The paper is well-written and organized, making it easy to follow.
+ The exploration of Linear Mode Connectivity within CIL is thought-provoking and provides valuable insights.
+ Extensive experiments on various datasets validate the method’s effectiveness.

**Weaknesses:**

+ The novelty of the proposed approach could be questioned, as similar formulations to the core result (Eq. 11) have appeared in prior works [1, 2, 3]. While the author’s starting point and implementation differ, it would be helpful to clarify essential distinctions between this work and related approaches.
+ The performance gains of IVT seem modest, particularly when applied to AFC.
+ The paper lacks direct comparisons with related methods, at least the classic baseline method [1].
+ The term "oracle" could benefit from a clearer definition. While readers may infer the meaning, it could cause some confusion. Maybe referring to it as the global optimum across all tasks, as suggested in footnote 1, might enhance clarity.

[1] Lee S W, Kim J H, Jun J, et al. Overcoming catastrophic forgetting by incremental moment matching[J]. Advances in neural information processing systems, 2017, 30.

[2] Sun W, Li Q, Zhang S, et al. Incremental Learning via Robust Parameter Posterior Fusion[C]//ACM Multimedia 2024.

[3] Matena M S, Raffel C A. Merging models with fisher-weighted averaging[J]. Advances in Neural Information Processing Systems, 2022, 35: 17703-17716.

**Questions:**

See weakness.

---

### Official Review · Reviewer_9Bxp · 2024-11-01

**Soundness:** 3
**Presentation:** 2
**Contribution:** 2
**Rating:** 5
**Confidence:** 5

**Summary:**

The authors introduce a method called Increment Vector Transformation (IVT) for Class Incremental Learning (CIL), which leverages Linear Mode Connectivity (LMC) to find low-loss paths for CIL models. IVT approximates the Hessian using the diagonal of the Fisher Information Matrix, balancing computational efficiency with the preservation of essential curvature information. The authors give an analysis based on the experimental results and theoretical analysis and introduce the IVT approach. Experimental results on benchmark datasets (CIFAR-100, ImageNet-Subset, and ImageNet-Full) demonstrate that integrating IVT with existing CIL methods leads to significant performance improvements.

**Strengths:**

- The motivation that follows a low-loss linear path to update the model seems interesting and effective.
- The IVT can identify low-loss paths without any exemplars, which is an advantage over EOPC.

**Weaknesses:**

- The image seems confusing, and I didn't understand what Fig. 2 meant at all until I read the discussion about Fig. 4, which gave me a rough idea of what the figure is trying to convey. However, I still don't fully understand why the $\lambda$ value is so large here, and what the decrease in accuracy in the middle signifies, and the Fig.3 and Fig.5 also suffer this. The LT in the captions of Fig.3 and Fig.5 should be LF.
- The performance seems not so competitive compared with EOPC, although the AA and LA outperform EOPC, the FM is obviously higher than EOPC as shown in Tab.1 and Tab.3.
- There should be more recent approaches for comparison. It's better to utilize IVT based on more approaches besides PODNet and AFC to show the robustness of the IVT.

**Questions:**

- Can there be a clearer explanation about the figures?
- Can IVT be utilized based on more other approaches?

---

### Official Review · Reviewer_qTv6 · 2024-11-04

**Soundness:** 3
**Presentation:** 4
**Contribution:** 3
**Rating:** 5
**Confidence:** 3

**Summary:**

This paper presents an in-depth experimental analysis of Linear Mode Connectivity in Class Incremental Learning (CIL) oracles and unveils a novel Increment Vector Transformation (IVT) module designed for plug-and-play integration. The methodology used to illustrate how models can retain proficiency in previously learned tasks by navigating low-loss linear paths between modes is both original and thoughtfully executed. Additionally, the application of the IVT module across established CIL methods like PODNet and AFC, leading to consistent performance enhancements, is noteworthy. However, the validation of IVT's effectiveness being limited to just two methods casts some uncertainty on its universal applicability.

**Strengths:**

The exploration of Linear Mode Connectivity within CIL oracles is a captivating and intellectually stimulating subject.

The document is well-crafted, with its clarity and rigorousness making it both comprehensible and educational.

The IVT module, with its flexible plug-and-play design, shows potential for broad adaptation across various CIL strategies.

**Weaknesses:**

The experimental focus solely on the diagonal elements of the Hessian matrix, neglecting the inter-parameter correlations, might limit the thoroughness of the theoretical analysis.

The verification of the IVT module's impact on only two distinct CIL methods leaves its widespread effectiveness as a plug-and-play solution unverified, raising questions about its general utility.

**Questions:**

The formulation of equation (4) appears to complicate the connection between lambda and the magnitude of (\theta_t-\theta_i^). Would a simpler expression like (1-\lambda)\theta_i^ + \lambda\theta_t not more directly illustrate the linear path as lambda varies from 0 to 1? Such a change could provide more straightforward insights into IVT's workings.

The choice to integrate IVT with PODNet and AFC specifically prompts curiosity regarding the module's broader suitability. What aspects of these methods make them particularly amenable to IVT? Additionally, extending IVT's evaluation to encompass a wider array of, possibly more sophisticated, CIL methods would greatly strengthen the argument for IVT's adaptability and broad relevance.

---

### Note · Authors · 2024-11-14

I have read and agree with the venue's withdrawal policy on behalf of myself and my co-authors.